# *FLT3*-ITD Measurable Residual Disease Monitoring in Acute Myeloid Leukemia Using Next-Generation Sequencing

**DOI:** 10.3390/cancers14246121

**Published:** 2022-12-12

**Authors:** Jong-Mi Lee, Silvia Park, Insik Hwang, Dain Kang, Byung Sik Cho, Hee-Je Kim, Ari Ahn, Myungshin Kim, Yonggoo Kim

**Affiliations:** 1Department of Laboratory Medicine, Seoul St. Mary’s Hospital, College of Medicine, The Catholic University of Korea, Seoul 06591, Republic of Korea; 2Catholic Genetic Laboratory Center, Seoul St. Mary’s Hospital, College of Medicine, The Catholic University of Korea, Seoul 06591, Republic of Korea; 3Department of Hematology, Seoul St. Mary’s Hospital, College of Medicine, The Catholic University of Korea, Seoul 06591, Republic of Korea; 4Leukemia Research Institute, Seoul St. Mary’s Hospital, College of Medicine, The Catholic University of Korea, Seoul 06591, Republic of Korea; 5Dow Biomedica Inc., Seoul 05771, Republic of Korea

**Keywords:** *FLT3*-ITD, next generation sequencing, measurable residual disease, acute myeloid leukemia, bone marrow transplantation, hematopoietic stem cell transplantation

## Abstract

**Simple Summary:**

*FLT3*-ITD monitoring is essential in AML management, and there is a great need for sensitive monitoring methods. We present a simple and easily applicable ITD-tracing algorithm optimized for MRD monitoring based on the NGS method. Our assay is sensitive to 0.001% and has a superior performance over conventional fragment analysis. For AML patients undergoing allo-HSCT, our assay showed that the MRD assessed before and after HSCT were significantly associated with a risk of relapse and a poor overall survival, respectively. This report highlighted the prognostic value of serial MRD monitoring using a sensitive method in a clinical setting of AML patients with *FLT3*-ITD.

**Abstract:**

The in-frame internal tandem duplication (ITD) of the FMS-like tyrosine kinase 3 (*FLT3*) gene is an important negative prognostic marker in acute myeloid leukemia (AML). *FLT3*-ITD monitoring is essential for patients at relapse or those receiving *FLT3*-targeted therapies. Fragment analysis (FA) is commonly used to detect and quantify *FLT3*-ITDs; however, detecting low-burden *FLT3*-ITDs after a treatment is challenging. We, therefore, developed a customized, next-generation sequencing (NGS)-based *FLT3*-ITD assay that includes a new ITD-tracing algorithm, “SEED”, optimized for measurable residual disease (MRD) monitoring. NGS-SEED showed an enhanced sensitivity (0.001%) and has a superior performance over conventional fragment analysis. We further investigated the prognostic impact of MRD analyzed by NGS-SEED in AML patients who underwent allogeneic hematopoietic stem cell transplantation (HSCT). Our assay showed that the MRD assessed before and after HSCT were significantly associated with a risk of relapse and a poor overall survival, respectively, in a time-dependent analysis. Thus, this report highlighted the prognostic value of serial MRD monitoring using a sensitive method in a clinical setting of AML patients with *FLT3*-ITD.

## 1. Introduction

The in-frame internal tandem duplications (ITDs) of the gene encoding FMS-like tyrosine kinase 3 (*FLT3*) have been found in approximately 25% of acute myeloid leukemia (AML) cases [1]. Approximately 70% of these duplications are located in the juxtamembrane domain and consistently increase the *FLT3* tyrosine kinase activity. *FLT3*-ITDs with allelic ratios ≥ 0.5 have been associated with an adverse prognosis [2,3]. In addition, *FLT3*-ITD monitoring is essential for patients at relapse and for those receiving *FLT3*-targeted therapy [4,5,6]. Fragment analysis (FA) using capillary electrophoresis has been the standard for detecting and quantifying *FLT3*-ITDs [7]. This assay has a high clinical utility because it provides reliable, rapid results. However, the detection of lower-burden *FLT3*-ITDs (<2%) after induction chemotherapy or hematopoietic stem cell transplantation (HSCT) is hampered by PCR bias [7,8,9].

Significant progress has been made in applying next-generation sequencing (NGS) to the detection and monitoring of *FLT3*-ITD [10,11,12,13,14], However, the detection of *FLT3*-ITD by NGS has been hindered by its unstable nature and complexity of the *FLT3*-ITD. The precise quantification of ITD burdens by NGS remains challenging because of the inherent biases and limitations of the standard clinical NGS pipelines. Multiple bioinformatics tools have been developed, but their performance is suboptimal for use without an adjustment [15,16].

To address these issues, we have developed a customized NGS-based *FLT3*-ITD assay optimized for the monitoring of measurable residual disease (MRD) with an enhanced sensitivity and specificity. We carefully evaluated the analytical performance of the NGS-based *FLT3*-ITD assay and applied it to track residual disease using clinical samples under morphological remission pre- and post-HSCT. We further described the clinical impact of the monitoring of MRD using the NGS-based *FLT3*-ITD assay.

## 2. Patients and Methods

### 2.1. Samples for Performance Evaluation

This study included clinical samples from 46 *FLT3*-ITD positive acute leukemia patients (45 with AML and one with T/myeloid mixed phenotype acute leukemia) who were diagnosed and received allo-HSCT at the Catholic Hematology Hospital from April 2013 to July 2020 (Table 1). A total of 211 BM samples were selected for the NGS-based *FLT3*-ITD assay, including those at diagnosis (*n* = 46), pre-HSCT (*n* = 79), post-HSCT at 1 month (*n* = 37), 3 months (*n* = 24), and 6 months (*n* = 25). Negative control samples were prepared from 10 patients with AML who were *FLT3*-ITD negative by FA (Appendix A). To evaluate the analytical performance of the NGS-based *FLT3*-ITD assay, we used 30 bp *FLT3*-ITD containing reference DNA (IVS-0017 Polyclonal Control DNA, InvivoScribe, Technologies, San Diego, CA), five clinical samples containing various sizes of ITDs (39, 78, 108, 156, and 206 bp, Appendix A), and their products (*n* = 74) serially diluted using *FLT3* wild-type DNA (IVS-0000 Clonal Control DNA, InvivoScribe, Technologies, San Diego, CA, USA).

### 2.2. Patients

Of the 46 patients with acute leukemia and *FLT3*-ITD, 42 received intensive chemotherapy to induce remission. Thirty-four patients (73.9%) were treated according to the standard protocol, which consisted of 3 + 7 idarubicin (IDA) and cytarabine (Ara-C). Seven patients (15.2%) were treated with 3 + 7 daunorubicin plus Ara-C. One patient with acute promyelocytic leukemia received all-trans retinoic acid (ATRA) plus IDA. Decitabine was used as an initial treatment regimen in two patients of an advanced age (67 and 68 years). Low-dose Ara-C plus etoposide was administered to one patient (64 years) due to infectious complications. One older patient (71 years) with multiple comorbidities participated in the ASP2215 study to receive gilteritinib as an induction regimen. Eleven (23.9%) adult patients received *FLT3* inhibitors as salvage chemotherapy (*n* = 5, quizartinib via expanded access program; *n* = 5, azacitidine + sorafenib; *n* = 1 gilteritinib) or induction chemotherapy (*n* = 1 gilteritinib via ASP2215 study). Two pediatric patients were treated with sorafenib in combination with IDA/Ara-C at relapse after the HSCT (Appendix A). Nine patients were enrolled in randomized, placebo-controlled clinical trials for the *FLT3* inhibitors; therefore, they were excluded from a further survival analysis in this specific variable. This study was carried out in accordance with the Declaration of Helsinki and was approved by the Institutional Review Board (IRB) of Seoul St. Mary’s Hospital (KC22RISI0206).

### 2.3. NGS-Based FLT3-ITD Assay

We developed an NGS-based *FLT3*-ITD assay to detect *FLT3*-ITD and optimized the bioinformatics pipeline to monitor MRD. The primer set covering *FLT3* exons 14–15 (forward-chr13:28,608,315-28,608,296, reverse-chr13:28,608,129-28,608,108) was designed to produce a 208 bp unique amplicon. The library was prepared using at least 50 ng of gDNA, which was measured using a Nanodrop spectrophotometer (Nanodrop, Wilmington, NC, USA). Paired-end sequencing reads (150 bp) were generated using a NextSeq sequencer (Illumina, San Diego, CA, USA). The 300 bp-sized long paired reads were sequenced using the MiSeq platform (Illumina, San Diego, CA, USA) to analyze longer ITDs 17. Negative results were confirmed using 400 ng of gDNA. The average depth of coverage was 829,568 per sample, and the minimum depth was 157,123.

### 2.4. SEED Algorithm for MRD of FLT3-ITD

Most ITDs typically have a short nucleotide sequence, termed a spacer, between the duplication and its origin [17]. Therefore, the entire insertion segment is composed of a spacer and duplication. We assigned a unique SEED sequence for each ITD, encompassing the junctional region of insertion and its origin, based on the structural features. The entire procedure of the sequencing data, including the alignment, recalibration, and SEED algorithm, is illustrated in Appendix A. We hypothesized that the SEED sequence would reflect the true allelic burden more precisely than the full ITD sequence. The size of the SEED sequence was set to 12–20 bp to optimize the detection sensitivity and specificity. To designate the SEED sequence, a Binary Alignment Map (BAM) file was visualized in the Integrative Genomics View (IGV) for the inspection of the ITD. Insertion or soft-clipped bases that showed >3 reads of identical sequences out of >10,000 reads were considered to be true ITDs. The variant allele frequency (VAF) values were calculated as the proportion of SEED-containing reads to the total read count. The command script that utilizes SAMtools to count the SEED reads from a BAM file is as follows:

> samtools view [bam.file] | grep GAAATCAACGGGAAACTCC | wc –l

The results were compared with those from FA and those analyzed by open-source tools for the detection of *FLT3*-ITD, including ITDseek (version 1.2, https://github.com/tommyau/itdseek (accessed on 26 August 2020)) [18], getITD (version 1.5.13, https://github.com/tjblaette/getitd (accessed on 26 April 2021)) [19], and Pindel (version 0.2.5b8, https://github.com/genome/pindel (accessed on 26 August 2020)) [20].

### 2.5. Statistical Analysis

The detection limit was determined to be 95% probit using probit regression analysis. The MRD positivity of *FLT3*-ITD was only determined for the samples at morphological remission (BM blasts < 5%). When multiple ITDs were detected in one sample, the sum of the ITDs was used. If multiple follow-up tests were performed at the same time point, the highest VAF value was selected to determine the MRD level. The MRD status at remission, combined with the clinical data, was examined for the prognostic value. We analyzed the predictive power of clinical and mutational factors for relapse, non-relapse mortality (NRM), event-free survival (EFS), and overall survival (OS). We defined EFS as the time from the HSCT to relapse or death, and OS as the time from the HSCT to death from any cause. The medical records were tracked until September 2021. The EFS and OS were estimated using the Kaplan–Meier method. A competing risk analysis was performed to estimate the probability of a cumulative incidence of relapse (CIR) and NRM. The CIR was compared across groups using the Gray test and cmprsk module in R [21,22]. The EFS and OS were compared using the Cox proportional hazards regression. The post-HSCT MRD and GVHD status at different time points were evaluated as the time-dependent covariates. Receiver-Operating Characteristic (ROC) analysis was performed to determine the optimal cut-off values for predicting relapse, ESF, and OS. For multivariate analysis, adverse variables with a *p*-value of <0.05 in the preceding univariate analysis were entered into a proportional hazard model for the sub-distribution of a competing risk or Cox proportional hazard model. Then, we fitted a joint model of longitudinal and time-to-event data to account for the informative dropout of the MRD measurements at 4 longitudinal time points (pre-HSCT, post-HSCT at 1 month, 3 months, and 6 months) due to relapse using the R package JM [23,24]. Statistical analyses were performed using MedCalc version 19.1.7 (MedCalc Software, Ostend, Belgium), SAS version 9.4 (SAS Institute Inc., Cary, NC, USA), and R software version 4.1.3 (R Foundation for Statistical Computing, Vienna, Austria).

## 3. Results

### 3.1. Assay Performance of NGS with SEED Algorithm for FLT3-ITD

The NGS-SEED showed an excellent sensitivity (up to 0.001%) and linearity (R^2^ = 0.991, *p* < 0.001) in the reference DNA analysis (Figure 1A). An additional evaluation of 84 diluted clinical samples demonstrated that the NGS-SEED identified most *FLT3*-ITDs (95.9%, 71/74), while other assays identified fewer: Pindel (93.2%, 69/74), getITD (66.2%, 49/74), and ITDseek (29.7%, 22/74) (Appendix A). The linearity was significant for the total data set (R^2^ = 0.853), but also for each ITD (Figure 1B; Appendix A). ITDs were not detected in 10 AML samples with wild-type *FLT3*, indicating a 100% specificity. These results collectively indicated that the limit of detection of the assay was 0.006% for a 95% probability (95% CI: 0.004–0.035%).

We then compared the results from the NGS-SEED with those from the FA for 211 clinical samples. The NGS-SEED detected 126 *FLT3*-ITDs in 114 samples, while the FA detected only 60 ITDs in 55 samples (Figure 1C). Fifty-three types of ITD were identified, ranging from 18 to 206 bp (median 48 bp), positioned between chr13:28,608,102 and chr13:28,608,313 (Appendix A). Seven patients had two ITDs, one of which emerged during the follow-up and was not detected by the FA due to its very low burden (VAF, 0.014%). We found another interesting case with two different ITDs that were not distinguished by the FA owing to their similar sizes (24 and 27 bp). The quantitative detection of *FLT3*-ITD correlated well between the NGS-SEED and FA, with a coefficient of 0.92 (*p* < 0.001, 95% CI: 0.90–0.94, Figure 1D). The VAFs from the NGS-SEED were lower than those from the FA (*p* < 0.001, Figure 1E).

We compared the detection capability of other bioinformatic tools against 50 types of ITDs accurately assigned by FA. Pindel detected ITDs in 98% (49/50) of cases, but their positions were misread in six cases (12.2%, 6/49). The ITDseek and getITD identified 84% (42/50) and 74% (37/50) of ITDs, respectively (Appendix A). The correlation coefficients of the VAF between the FA and other bioinformatic tools were as follows: NGS-SEED (R = 0.841, *p* < 0.001), Pindel (R = 0.807, *p* < 0.001), getITD (R = 0.684, *p* < 0.001), and ITDseek (R = 0.451, *p* = 0.003) (Appendix A).

### 3.2. FLT3-ITD Assessment after Treatment

The *FLT3*-ITD was assessed pre-HSCT (*n* = 35), post-HSCT-1 month (*n* = 36), 3 months (*n* = 19), and 6 months (*n* = 12) in patients with morphological remission. Twenty-eight of the 35 patients (80%) harbored persistent pre-HSCT MRD. Among them, 4 (11.4%) patients showed MRD which was detected by both the FA and NGS-SEED (median VAF: 5%), whereas 24 (68.6%) patients showed the MRD detected only by the NGS-SEED (median VAF: 0.048%). The MRD values measured by the NGS-SEED alone were as follows: 9 (25.7%) in MRD > 0.1%, 8 (22.9%) in MRD 0.1–0.01%, and 7 (20.0%) in MRD 0.01–0.001% with a median VAF of 0.375%, 0.041%, and 0.009%, respectively. The post-HSCT MRD was identified by the NGS-SEED in 6 (16.7%) with a median VAF of 0.19% at 1 month, and post-HSCT at 3 months and 6 months. MRD was detected in three (15.8%) and four (33.3%) patients with median VAF values of 0.01% and 0.15%, respectively. None of these were detected by the FA (<2%) (Figure 2).

### 3.3. Prognostic Impact of FLT3-ITD MRD by NGS-SEED

We further investigated the prognostic impact of the *FLT3*-ITD MRD before and after the HSCT. The median follow-up duration was 37 months (95% CI: 33–104), and the 3-year OS was 54.5% ± 9.9%. The median EFS was 28 months (95% CI: 18–28) after excluding a patient who died of graft failure 87 days following the HSCT. The 3-year EFS was 50.0% (95% CI: 36.2–65.5%). The 3-year overall cumulative incidence of relapse (CIR) and NRM was 29.1% (95% CI: 16.7–42.6%) and 20.8% (95% CI: 10.2–34.0%), respectively.

Detailed prognostic data are presented in Table 2. In the univariate analysis, pre-HSCT MRD was a significant risk factor for relapse by 0.1%, 0.01%, and 0.001% cut-offs. (Appendix A). Post-HSCT MRD > 0.1% and >0.001% were also associated with relapse (Appendix A). Post-HSCT MRD > 0.1% and >0.001% were associated with a poor EFS. An inferior OS was only associated with a post-HSCT MRD > 0.1% (Appendix A).

In the ROC curve analysis, pre-HSCT MRD > 0.1%, and post-HSCT MRD > 0.001% showed the highest area under the curve (AUC) for predicting relapse (AUC = 0.730, 0.747, respectively), and post-HSCT MRD > 0.001% and > 0.1% were the optimal classifiers for EFS and OS, respectively (AUC = 0.661, 0.672, respectively) (Appendix A). Then, we performed a multivariate analysis using the optimal MRD factors and potential confounders. As shown in Table 3, pre-HSCT MRD > 0.1% was significantly associated with relapse, while post-HSCT MRD > 0.1% was the only significant predictor for the OS.

## 4. Discussion

MRD is one of the most powerful independent prognostic markers for AML [13,25,26,27]. With the emergence of targeted agents, the role of the monitoring of MRD in the management of AML is growing. The recent guidelines recommend the use of MRD with a detection limit lower than 0.1% [13,26,28]. For *FLT3*-ITD, the current recommended standard method for detection and quantification is FA, but its sensitivity is challenging, at approximately 2% [9]. Recently, multiple NGS-based assays for the assessment of *FLT3*-ITD have been introduced, but they still require an adjustment for a clinical implementation owing to their limited analytical performance [15,16,17,29].

In the current study, we developed a novel assay using the SEED sequence to analyze the *FLT3*-ITD MRD using the NGS data. In our assay, the initial ITD was searched using the IGV window, and thereafter, a specific SEED sequence to the ITD was assigned and further queried from the entire sequencing data. This strategy improved the analytical accuracy by preserving the mutated SEED sequence reads, even if there were polymorphisms and/or random sequencing errors.

We have shown that the NGS-SEED detects all types of ITDs, including long ITDs, when other assays frequently misidentify or misread their positions. In our analysis of 211 clinical samples, low-burden mutations were additionally detected using NGS-SEED in 59 FA-negative samples. The NGS-SEED showed an excellent linearity across ITDs of various sizes and achieved a detection limit of approximately 0.001%. Tracing the ITD-specific sequence of SEED was also advantageous for distinguishing separate clones with similar ITD sizes. Moreover, the NGS-SEED algorithm can be applied to other *FLT3*-ITD NGS datasets using other primer sets.

Using technical accomplishments, we further investigated the prognostic impact of MRD according to various cut-offs and time points. The NGS-SEED had a significant impact in both pre- and post-HSCT. We monitored the MRD pre-HSCT and post-HSCT at 1, 3, and 6 months. Eighty percent of our patients harbored persistent pre-HSCT MRD, and 86% of them were detected by NGS-SEED. It appeared that the best pre-HSCT MRD cut-off to predict relapse was 0.1%, with 37% of patients showing an MRD above it. The significance of pre-HSCT *FLT3*-ITD MRD as a risk factor for relapse have been shown by other groups using NGS methods. Hourigan CS et al. found that 9 out of 10 patients with *FLT3*-ITD MRD with VAF 0.03–0.97% relapsed after HSCT, while 1 out of 7 patients with undetectable MRD relapsed [30]. The recent study by Loo et al. described the prognostic impact of *FLT3*-ITD MRD at a 0.001% cut-off, which is much lower than our result [31]. The optimal NGS-MRD cut-off for the relapse prediction has not yet been defined, so a further investigation is required.

After allo-HSCT, the *FLT3*-ITD MRD was decreased to a measurable level by NGS alone. As a time-dependent covariate, post-HSCT MRD > 0.1% was significantly associated with poor OS in the multivariate analysis. In addition, patients with post-HSCT MRD had a significantly higher CIR rate. In the subsequent analysis using the joint model of longitudinal and time-to-event data, we could validate the prognostic impact of the longitudinal MRD value, which has been rarely described before [32].

These results collectively indicate that NGS-based *FLT3*-ITD MRD assays should be considered for serial use in pre- and post-HSCT [33]. Considering that the sensitivity of NGS-based *FLT3*-ITD assays in this and previous studies was nearly 0.001–0.005% [19,34], it is relevant to obtain an accurate MRD detection near the significant cut-off.

This study had several limitations. Given the retrospective nature of our analysis, we could not control for the considerable heterogeneity in patient characteristics, disease status, and type of treatment. In addition, the statistical power of this study was limited by a small sample size. Therefore, our conclusions should be further confirmed in a lager prospective cohort. The manual ITD searching process of NGS-SEED requires further improvement before it can be widely applied. A dedicated user interphase and software is now underdevelopment to enhance the reproducibility and shorten the turn-around time of this method. Despite these limitations, this study clarified the significance of pre-HSCT *FLT3*-ITD MRD as a risk factor for relapse. In addition, post-HSCT MRD is a strong risk factor for a poor survival.

## 5. Conclusions

In this study, we demonstrated the prognostic impact of the sensitive *FLT3*-ITD MRD using SEED-NGS. Then, we carefully proposed the optimal cut-off level for *FLT3*-ITD MRD in patients with AML who underwent allo-HSCT according to various time points. This report also highlighted the prognostic value of serial MRD monitoring using a sensitive method in a clinical setting of AML patients with *FLT3*-ITD. Further studies incorporating targeted therapies in large cohorts of patients may potentiate the significance of *FLT3*-ITD MRD and broaden its application.

## Figures and Tables

**Figure 1 cancers-14-06121-f001:**
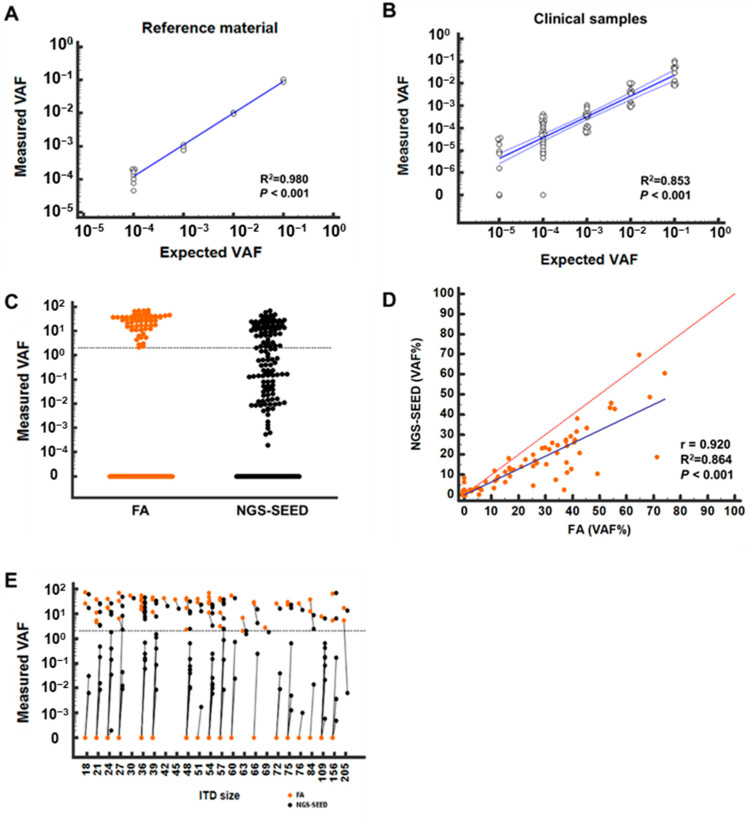
Analytical performance of NGS-SEED: (**A**) linearity validation using reference material with a 30-bp ITD. The blue line indicates regression line; (**B**) linearity validation using clinical samples containing various ITDs of 39, 78, 108, 156, and 206 bp. Blue, regression line; blue dashed lines, 95% confidence interval; (**C**) variant allele frequencies (VAFs) measured by fragment analysis (FA) and NGS-SEED. Dashed line, detection limit of FA (2%); (**D**) VAF correlations between FA and NGS-SEED analysis of clinical samples at diagnosis and during follow-up. Red, equality line; blue, regression line; (**E**) paired VAFs measured using FA and NGS-SEED. NGS-SEED identified low-burden ITDs but underestimated high-burden ITDs.

**Figure 2 cancers-14-06121-f002:**
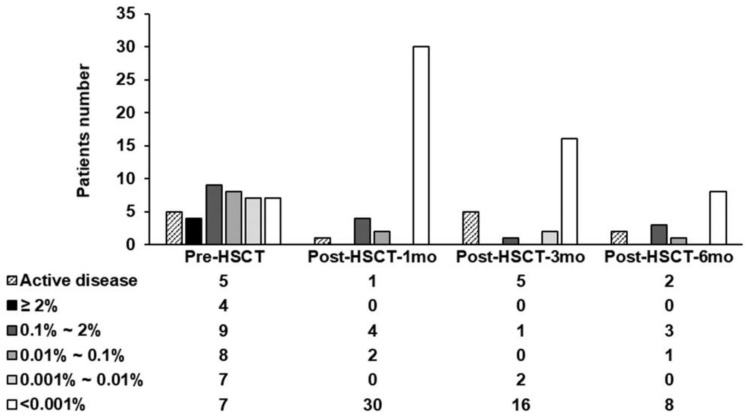
Patients for *FLT3*-ITD MRD analysis. Patients at active disease and measurable residual disease status of the patients at each time points.

**Table 1 cancers-14-06121-t001:** Patient demographics.

Characteristics		Total (*n* = 46)	MRD Assessed Patients *			
			Pre-HSCT	Post-HSCT	Post-HSCT	Post-HSCT
			(*n* = 35)	1 mo (*n* = 36)	3 mo (*n* = 19)	6 mo (*n* = 12)
Age at diagnosis (years)		43.3 ± 16.6	43.5 ± 16.1	46.5 ± 17.0	40.8 ± 18.5	34.3 ± 18.6
Gender (Male)		19 (41.3%)	17 (48.6%)	11 (30.6%)	7 (36.8%)	7 (58.3%)
WBC at diagnosis (×10^9^/L)		52.5 (19.5–110.4)	46.0 (16.5–121.4)	38.5 (8.3–103.6)	61.9 (2.9–14.6)	58.1 (9.1–204.6)
Initial ITD maximum length		46 (35.9–52.3)	45 (35.0–51.3)	45.5 (35.0–53.0)	41.0 (25.6–53.3)	43.5 (30.6–63.4)
Disease type	De novo AML	37 (80.4%)	28 (80.0%)	28 (77.8%)	14 (73.7%)	9 (75.0%)
	AML-MRC	7 (15.2%)	5 (14.3%)	6 (16.7%)	3 (15.8%)	2 (16.7%)
	Therapy-related AML	1 (2.2%)	1 (2.9%)	1 (2.8%)	0 (0%)	0 (0%)
	MPAL (T/M)	1 (2.2%)	1 (2.9%)	1 (2.8%)	1 (5.3%)	1 (8.3%)
Mutations	*NPM1*	17 (37.0%)	14 (40.0%)	13 (36.1%)	7 (36.85)	5 (41.7%)
	*CEBPA*	3 (6.5%)	3 (8.6%)	2 (5.6%)	1 (5.3%)	1 (8.3%)
2017 ELN risk group	Favorable	14 (30.4%)	14 (40.0%)	12 (33.3%)	6 (31.6%)	3 (25.0)
	Intermediate	12 (26.1%)	8 (22.9%)	7 (19.4%)	4 (21.1%)	4 (33.3%)
	Poor	20 (43.5%)	13 (37.1%)	17 (47.2%)	9 (47.4%)	5 (41.7%)
Initial induction regimen	Intensive chemotherapy	42 (91.3%)	32 (91.4%)	33 (91.7%)	16 (84.2%)	12 (100%)
	Low-intensive chemotherapy	3 (6.5%)	3 (8.6%)	2 (5.65)	1 (5.3%)	0 (0%)
	*FLT3* inhibitor	1 (2.2%)	0 (0%)	1 (2.8%)	1 (5.3%)	0 (0%)
Chemotherapy cycles before CR	1 cycle	25 (54.3%)	24 (68.6%)	17 (47.2%)	11 (57.9%)	6 (50.0%)
	2 cycles	11 (23.9%)	10 (28.6%)	10 (27.8%)	4 (21.1%)	2 (16.7%)
	3 cycles	1 (2.2%)	1 (2.9%)	1 (2.8%)	1 (5.3%)	0 (0%)
Conditioning intensity	Myeloablative	37 (80.4%)	30 (85.7%)	32 (88.9%)	15 (78.9%)	10 (83.3%)
	Reduced intensity	9 (19.6%)	5 (14.3%)	4 (11.1%)	4 (21.1%)	2 (16.7%)
*FLT3* inhibitor	No	23 (50.0%)	20 (57.1%)	18 (50.0%)	8 (42.1%)	4 (33.3%)
	Yes	14 (30.4%)	8 (22.9%)	13 (36.1%)	7 (36.8%)	5 (41.7%)
	Clinical trial	9 (19.6%)	7 (20.0%)	5 (13.9%)	4 (21.1%)	3 (25.0)
Disease status at HSCT	CR1	34 (73.9%)	32 (91.4%)	27 (75.0%)	16 (%)	7 (58.3%)
	CR2	4 (8.7%)	3 (8.6%)	2 (5.65)	0 (0%)	3 (25.0%)
	Persistent leukemia	8 (17.4%)	0 (0%)	7 (19.4%)	3 (15.8%)	2 (16.7%)
Transplant modality	Matched sibling donor	15 (32.6%)	13 (37.1%)	11 (30.6%)	8 (42.1%)	5 (41.7%)
	Matched unrelated donor	13 (28.3%)	9 (25.7%)	10 (27.8%)	4 (21.1%)	2 (16.7%)
	Haploidentical donor	18 (39.1%)	13 (37.1%)	15 (41.7%)	7 (36.8%)	5 (41.7%)
GVHD	Acute GVHD	22 (66.7%)	-	13 (48.1%)	5 (33.3%)	5 (55.6%)
	Chronic GVHD	14 (42.4%)	-	9 (33.3%)	3 (20%)	2 (22.2%)
Clinical outcome	Died without relapse	9 (19.6%)	7 (20.0%)	8 (22.2%)	3 (15.8%)	0 (0%)
	Relapsed and died	12 (26.1%)	9 (25.7%)	9 (25.0%)	3 (15.8%)	3 (25.0)
	Died with persistent leukemia	1 (2.2%)	0 (0%)	0 (0%)	0 (0%)	0 (0%)
	Ongoing CR after reinduction chemotherapy for relapse	1 (2.2%)	1 (2.9%)	1 (2.8%)	1 (5.3%)	1 (8.3%)
	Ongoing CR	23 (50.0%)	18 (51.4%)	18 (50.0%)	12 (63.2%)	8 (66.7%)

* MRD assessed patient: patients with morphological CR at the time of testing. Abbreviations: WBC, white blood cells; ITD, internal tandem duplication; AML, acute myeloid leukemia; MRC, myelodysplasia-related changes; MPAL, mixed phenotype acute leukemia; ELN, European LeukemiaNet; HSCT, hematopoietic stem cell transplantation; CR, complete remission; CR1, first complete remission; CR2, second complete remission; GVHD, graft-versus-host disease; MRD, minimal residual disease; 1mo, 1 month; 3mo, 3 months; 6mo, 6 months.

**Table 2 cancers-14-06121-t002:** Univariate analysis.

Univariate Variables	*n*	Cumulative Incidence of Relapse		Cumulative Incidence of Non-Relapse Mortality		Event Free Survival		Overall Survival	
		HR (95% CI)	*p* Value	HR (95% CI)	*p* Value	HR (95% CI)	*p* Value	HR (95% CI)	*p* Value
MRD status at pre-HSCT >2% >0.1% >0.01% >0.001%	35 4 13 21 28	2.79 (0.53–14.59) **5.49 (1.55–19.45)** **7.50 (1.01–55.69)** **1.3 × 10^7^ (5.25 × 10^6^–3.55 × 10^7^)**	0.226 **0.008** **0.049** **<0.001**	0.26 (0.03–1.98) 0.86 (0.20–3.75) 0.57 (0.11–3.07) NA	0.192 0.841 0.510 NA	1.22 (0.28–5.35) 1.80 (0.69–4.70) 2.65 (0.86–8.16) 1.99 (0.45–8.74)	0.794 0.230 0.090 0.364	1.25 (0.28–5.71) 1.48 (0.55–4.02) 2.50 (0.80–7.83) 1.87 (0.42–8.31)	0.767 0.439 0.116 0.411
MRD status at post-HSCT >0.1% >0.001%	44 6 10	- -	- -	- -	- -	**4.73 (1.88–11.93)** **2.86 (1.15–7.14)**	**0.001** **0.024**	**4.04 (1.61–10.18)** 2.38 (0.95–5.98)	**0.003** 0.065
Age group <60 years ≥60 years	45 38 7	1 1.68 (0.51–5.49)	0.392	1 2.88 (0.80–10.4)	0.104	1 2.37 (0.93–6.06)	0.072	1 2.52 (0.98–6.52)	0.057
Gender Female Male	45 26 19	1 2.33 (0.79–6.87)	0.125	1 0.34 (0.08–1.52)	0.157	1 1.07 (0.46–2.48)	0.870	1 0.86 (0.36–2.04)	0.727
WBC group at diagnosis <50 × 10^9^/L ≥50 × 10^9^/L	45 23 22	1 2.75 (0.89–8.49)	0.079	1 0.47 (0.12–1.83)	0.276	1 1.35 (0.58–3.12)	0.490	1 1.21 (0.48–2.65)	0.794
Initial ITD maximum length ˂50 bp ≥50 bp	45 27 18	1 0.87 (0.30–2.52)	0.794	1 2.04 (0.57–7.27)	0.272	1 1.29 (0.56–2.99)	0.552	1 1.40 (0.59–3.30)	0.442
2017 ELN risk group Favorable Intermediate Adverse	45 14 11 20	1 8.20 (0.94–71.80) 5.25 (0.65–42.60)	0.164 0.057 0.120	1 0.56 (0.12–2.74) 0.51 (0.11–2.25)	0.618 0.477 0.372	1 2.51 (0.73–8.63) 2.26 (0.72–7.13)	0.240 0.143 0.163	1 1.89 (0.53–6.69) 2.14 (0.68–6.74)	0.378 0.327 0.194
Chemotherapy cycles for CR 1 cycle ≥2 cycles	37 25 12	1 2.21 (0.69–7.09)	0.180	1 1.58 (0.38–6.59)	0.534	1 2.10 (0.82–5.35)	0.121	1 2.57 (0.97–6.76)	0.057
Conditioning intensity Myeloablative Reduced intensity	45 36 9	1 **2.83 (1.01–7.92)**	**0.048**	1 1.03 (0.25–4.28)	0.967	1 1.93 (0.79–4.76)	0.150	1 1.87 (0.75–4.64)	0.177
*FLT3* inhibitor No Yes	36 22 14	1 1.73 (0.60–5.00)	0.311	1 0.45 (0.10–2.04)	0.300	1 0.90 (0.37–2.21)	0.820	1 0.77 (0.30–1.95)	0.579
Disease state at HSCT CR1 CR2 Persistent leukemia	45 33 4 8	1 1.58 (0.39–6.33) NA	NA 0.519 NA	1 1.29 (0.16–10.40) 0.54 (0.07–4.04)	0.789 0.811 0.547	1 1.52 (0.44–5.22) 0.880 (0.26–3.02)	0.782 0.507 0.839	1 1.58 (0.46–5.45) 0.99 (0.29–3.44)	0.787 0.473 0.993
Transplant modality Matched sibling Matched unrelated Haploidentical	45 14 13 18	1 3.79 (0.85–17.01) 1.61 (0.29–8.96)	0.143 0.082 0.586	1 1.06 (0.22–5.00) 0.71 (0.15–3.33)	0.862 0.946 0.667	1 2.82 (0.70–11.31) **4.42 (1.25–15.57)**	**0.032** 0.143 **0.021**	1 2.63 (0.66–10.57) **3.87 (1.09–13.75)**	**0.065** 0.172 **0.036**
Acute GVHD None Grade ≥ 1	33 11 22	-	-	-	-	1 0.97 (0.35–2.71)	1 0.958	1 1.29 (0.44–3.76)	0.640
Chronic GVHD None Mild to Severe	33 19 14	-	-	-	-	1 0.60 (0.20–1.89)	0.382	1 0.68 (0.21–2.18)	0.517

Significant findings for adverse events are highlighted in bold. Abbreviations: n, number of patients; WBC, white blood cells; HR, hazard ratio; 95% CI, 95% confidential interval; ELN, European LeukemiaNet; HSCT, hematopoietic stem cell transplantation; CR, complete remission; CR1, first complete remission; CR2, second complete remission; GVHD, graft-versus-host disease; MRD, minimal residual disease; NA, not available.

**Table 3 cancers-14-06121-t003:** Muitivariate analysis.

Variables	*n*	Relapse		Event-Free Survival		Overall Survival	
		HR (95% CI)	*p* Value	HR (95% CI)	*p* Value	HR (95% CI)	*p* Value
MRD > 0.1% at pre-HSCT	13	**5.10 (1.53–17.02)**	**0.008**	-	-	-	-
MRD > 0.1% at post-HSCT	6	-	-	-	-	**2.61 (1.01–6.75)**	**0.049**
MRD > 0.001% at post-HSCT	10	-	-	1.90 (0.74–4.88)	0.185	-	-
Transplant modality = haploidentical	18	-	-	**4.72 (1.04–21.45)**	**0.044**	3.62 (0.77–16.96)	0.103
Conditioning intensity = reduced intensity	9	**3.24 (1.13–9.32)**	**0.029**	-	-	-	-

Significant findings are highlighted in bold Abbreviations: n, number of patients; HR, hazard ratio; 95% CI, 95% confidential interval; HSCT, hematopoietic stem cell transplantation; MRD, minimal residual disease; NI, not included in the multivariate model. In a joint model of longitudinal and time-to-event data, MRD was the reliable predictor for relapse (*p* = 0.002, hazard ratio: 1.17, 95% CI: 1.06–1.29), even after adjusting for a potential confounder; conditioning intensity (*p* = 0.564).

## Data Availability

The datasets generated during and/or analyzed during the current study are available in the NCBI SRA repository (BioProject ID:PRJNA880286).

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
