# Peer review of "FLT3-ITD Measurable Residual Disease Monitoring in Acute Myeloid Leukemia Using Next-Generation Sequencing"

_cancers, 2022, doi:10.3390/cancers14246121_

Round 1

Reviewer 1 Report

In this study, the authors developed a customized NGS-based FLT3-ITD assay for MRD monitoring in patients with AML undergoing allogeneic stem cell transplantation and showed that this method of MRD detection were significantly associated with risk of disease relapse post-transplant.  

The manuscript is well written. However, there are some critical issues regarding statistical analyses which could potentially impact the study result. Below please see my major comments.

1.     Please note that MRD status at different time points post-transplant are not available/known at the time of transplant; it has a time-dependent onset, so it is a time-dependent covariate not a time-fixed covariate. While time dependent covariate regression of OS and EFS with Cox model is feasible, time-dependent covariate in competing risk regression on relapse and NRM is problematic. Please see reference: ''A review of the use of time-varying covariates in the Fine-Gray subdistribution hazard competing risk regression model'', Statistics in Medicine. 2020;39:103–113.

For this reason, the impact of post-transplant MRD on cumulative incidence of relapse and NRM are not correctly calculated and results resented in this manuscript could be midleading.

For this type of data, joint models of longitudinal and time-to-event data with more than one event time outcome which accounts for informative dropout of the longitudinal measurements of post-transplant MRD due to death or relapse might be more appropriate.

2.     How do the authors explain the results of univariable analyses in table 3, when pre-transplant MRD > 2% was not associated with relapse while >0.01% significantly predicted the risk of relapse?

3.     In section 3.3, the authors stated that “The 3-year EFS was 53.1% ± 10.1%. The 3-year overall cumulative incidence of relapse and NRM was 29.8% (95% CI: 17.0-43.8%) and 20.6% (95% CI: 10.2-33.7%), respectively.” From the definition of EFS described in the method section. EFS+ Relapse+ NRM at the same time point should be 100%. Please check the accuracy of these numbers.

4.     Several important factors associated with post-transplant outcomes such as donor type, comorbidity index, performance status etc were not included in the univariable and multivariable analyses.

5.     The authors should comment on the sample size limitation in the discussion.

6.     The predictive performance of the MRD test on relapse as well as the other time to event outcomes should be reported using appropriate statistical test such a time-dependent ROC curve etc.

7.     How did the authors handle the penalty of multiple comparisons when the authors tested the impact of MRD using several cutoffs on several outcomes?

Reviewer 2 Report

NGS-SEED method was evaluated as a tool for MRD detection in FLT3-ITD positive AML in the patients undergoing allogeneic stem cell transplantation. I am curious why the name SEED? What is the difference from other NGS-based methods and especially from the PCR-NGS methods?

This method is not standardised and needs further evaluation and validation in another cohort of patients.

Furthermore, the group is small with 45 patients and heterogeneity of the patients is an obstacle to clinical conclusions. The patients were treated differently and there are also pediatric patients included. Are there any patients who got FLT3-inhibitors? Are there any difference in OS between MAC and RIC? 

I think that the clinical conclusions should be omitted from the abstracts and the authors should have more insisted on the methodological component of their work. It needs to be emphasised the manual aspect of the SEED method and it needs to be clarified the TAT and reproducibility of this approach. The optimal cut-off is still not known because there are no published prospective phase III studies.  

The authors conclude that NGS-SEEDs showed higher sensitivity and better performance than other NGS-based FLT3-ITD detection algorithms. What is the evidence of this in the paper? It needs to be clarified further. My suggestion is that the authors tone down these conclusions because the study aim is not the comparison of the different NGS-based methods and it is rather describing the outcome of a cohort of patients.  

Author Response

Point 1: NGS-SEED method was evaluated as a tool for MRD detection in FLT3-ITD positive AML in the patients undergoing allogeneic stem cell transplantation. I am curious why the name SEED? What is the difference from other NGS-based methods and especially from the PCR-NGS methods?

Response 1: SEED is not a new word, but is a bioinformatic term. We have adopted the concept of “SEED” from   modern alignment algorithms which utilize seed and extend strategy. In the seed and extend strategy, about 10bp sized short reads extracted from target read are assigned as seed sequence. Then the sequencing reads are extended from the seeds for alignment (Pharmaceutics. 2015 Dec; 7(4): 523–541). This strategy is advantageous because seed alignment is much easier and faster than alignment without it. NGS-SEED uses the concept of the seed in detection of ITD containing reads.

PCR-NGS describe by Loo S, et al, (Blood. 2022 Aug 12;blood.2022016567) amplifies wild-type FLT3 exon 14 and FLT3-ITDs by PCR before proceeding NGS. In the first step of PCR, one or two amplicons are generated depending on the ITD size and insertion site. Using the PCR products, NGS libraries are prepared and sequenced.

This assay has shown excellent performance in detection of small amount of FLT3-ITD coupled with get ITD (leukemia, 2019, 33:2535–2539) as an ITD detection algorithm. Despite its high sensitivity, this approach could be laborious and expensive, so it may not be available for everyone (Cancers 2022, 14, 4006). Our method has not adopted the primary PCR step but optimized the detection capacity using NGS-SEED algorithm. We believe that both assays have different strengths and weaknesses, but both are excellent tools for MRD monitoring in FLT3-ITD-positive AML.

Point 2: This method is not standardised and needs further evaluation and validation in another cohort of patients.

Furthermore, the group is small with 45 patients and heterogeneity of the patients is an obstacle to clinical conclusions. The patients were treated differently and there are also pediatric patients included. Are there any patients who got FLT3-inhibitors? Are there any difference in OS between MAC and RIC? 

Response 2: We totally agree with the reviewer’s point. Our study cohort included 14 patients who got FLT3 inhibitors and 23 patients who didn’t (Table 1). When we compared the clinical outcomes of the two treatment groups, there was no significant difference between them (Table 2). For conditioning intensity, RIC treated patients showed higher probability of relapse than MAC treated patients in univariate and multivariate analysis (Table 2,3). However, EFS and OS were not significantly different between them (Table 2). As you also pointed out, our study cohort is heterogeneous and small to completely validate the clinical impact of the treatment regimens. Therefore, we have added special comments in the discussion section about the limited sample sized and necessity of the further validation in line 320-321 as follow: The statistical power of this study was limited by small sample size. Therefore, our conclusions should be further confirmed in lager prospective cohort.

Point 3: I think that the clinical conclusions should be omitted from the abstracts and the authors should have more insisted on the methodological component of their work. It needs to be emphasised the manual aspect of the SEED method and it needs to be clarified the TAT and reproducibility of this approach. The optimal cut-off is still not known because there are no published prospective phase III studies.  

Response 3: Thank you for the direction and valuable suggestions. According to the reviewer’s comment, we have omitted the clinical validation part in summary and abstract. In addition, we added comments on the manual aspect in line 323-324 as follow: Dedicated user interphase and software is now underdevelopment to enhance reproducibility and shorten turn-around-time of this method.

Point 4: The authors conclude that NGS-SEEDs showed higher sensitivity and better performance than other NGS-based FLT3-ITD detection algorithms. What is the evidence of this in the paper? It needs to be clarified further. My suggestion is that the authors tone down these conclusions because the study aim is not the comparison of the different NGS-based methods and it is rather describing the outcome of a cohort of patients.  

Response 4: As you pointed out, the aim of study is to develop a sensitive and precise measurement tool for FLT3-ITD, not the comparison of the other methods. The evidence for the superiority of NGS-SEED to the other bioinformatics tools are based on detection sensitivity (100% by NGS-SEED, 98% by Pindel, 84% by ITDseek and 74% by get ITD) and VAF correlation to FA (R=0.84 by NGS-SEED, R=0.8 by Pindel, R=0.69 by get ITD and R=0.45 by ITDseek), as shown in the result 3.1. Pindel’s performance was comparable to NGS-SEED, but it sometimes misread the chromosomal position of the ITD. However, the algorithm performances could be influenced by various testing factors, such as primers design, DNA input, read depth, ITD types and so on. Your suggestion sounds reasonable, so we amended the abstract and discussion parts as you recommended.

We appreciate you for your precious time in reviewing our paper and providing valuable comments. It was your valuable and insightful comments that led to possible improvements in the current version. The authors have carefully considered the comments and tried our best to address every one of them. We hope the manuscript after careful revisions meet your high standards. In addition, we are planning to get another English editing after this revision. We look forward to hearing from you in due time regarding our submission and to respond to any further questions and comments you may have.

Reviewer 3 Report

Acute myeloid leukemia (AML) with Flt3 abnormalities represents an important proportion of AML patients and in spite of the emergence of various Flt3-inhibitors the management of this type of AML remains challenging. Thus the assessment of measurable residual disease (MRD) by FLT3-ITD monitoring is of the greatest importance and the use of more sensitive investigation methods is of great relevance.

The authors performed a retrospective study on clinical samples from 46 FLT3-ITD positive AML patients treated by different regimens over a period of 7 years in a single institution. The authors present a novel NGS-based method  for ITD detection and a improved ITD-tracing algorithm optimized for MRD monitoring. The results obtained on the study population showed higher sensitivity leading to better detection of ITD's compared to conventional fragment analysis method. Most important is the finding of the prognostic impact of  MRD according to different time-points and cutt-off values before and after HSCT suggesting the benefit of peritransplant serial MRD monitoring.

Despite its limitations (retrospective, limited and heterogenous patient population), the study introduces a novel, more performant NGS-based MRD monitoring method and highlights the prognostic value of serial MRD evaluation in the transplant setting for AML FLT3 positive patients.

I recommend publication in its present form

Author Response

Thank you for the encouraged and positive feedback.

Round 2

Reviewer 1 Report

The authors have appropriately addressed my comments and concerns.

However, there are some minor issues that need to be clarified.

1. What does "relapse and on going CR" in table 1 mean?

2. The impact of post-transplant MRD on cumulative incidence outcome (relapse and NRM) should not be presented in table 2 because HR of these variables (time-dependent) cannot be straightforwardly interpreted.

For the impact of post-transplant MRD, the authors should only focus on the results of joint model analysis.

3. Similarly, the impact of acute/chronic GVHD on survival outcomes should be used as time dependent variables (status is unknown at transplant). Moreover, their impact on CIR an NRM should not be analyzed using a simple competing risk analysis.

Author Response

The authors have appropriately addressed my comments and concerns.However, there are some minor issues that need to be clarified.

We want to thank you for the encouraging response and the opportunity to resubmit a revised manuscript. We have made every attempt to fully address your comments in this revised manuscript. Please find our response to your concerns below.

  1. What does "relapse and on going CR" in table 1 mean?

We are very sorry to cause this confusion. One patient who was alive in CR after reinduction chemotherapy for post-HSCT relapse. We have changed it to “Ongoing CR after reinduction chemotherapy for relapse.”

  1. The impact of post-transplant MRD on cumulative incidence outcome (relapse and NRM) should not be presented in table 2 because HR of these variables (time-dependent) cannot be straightforwardly interpreted.

For the impact of post-transplant MRD, the authors should only focus on the results of joint model analysis.

We agree with the reviewer’s point. According to the reviewer’s suggestion, we omitted the impact of post-HSCT MRD on cumulative incidence outcomes in table 2. We also amended the discussion in lines 309-314 to emphasize the result of the joint model analysis.

  1. Similarly, the impact of acute/chronic GVHD on survival outcomes should be used as time dependent variables (status is unknown at transplant). Moreover, their impact on CIR an NRM should not be analyzed using a simple competing risk analysis.

We sincerely appreciate this comment. Initial data was the worst grade of GVHD during the follow-up period. To address reviewer's suggestion, we collected the longitudinal GVHD data and reanalyzed them using the time-dependent Cox model for EFS and OS. However, the GVHD status of 12 patients was incomplete to obtain serial data. So we could perform the time-dependent Cox analysis for 33 available patients. As shown in table 2, they were not clinically significant. Unfortunately, we could not estimate the GVHD impact on CIR and NRM using joint modeling due to the limited sample size.

We appreciate the reviewer’s precious time in reviewing our paper and providing valuable comments. The authors have carefully considered the comments and tried our best to address every one of them. We hope the manuscript, after careful revisions meet the reviewer’s high standards. In addition, we are planning to get another English editing after this revision. We look forward to hearing from you in due time regarding our submission and to respond to any further questions and comments you may have.

Reviewer 2 Report

The authors changes in the manuscript has answered my concerns.  

Author Response

(The authors gave the same response as above.)
